# A DFT Study of the Reaction of Acrylamide with L-Cysteine and L-Glutathione

**DOI:** 10.3390/molecules27238220

**Published:** 2022-11-25

**Authors:** Sandra Ramirez-Montes, Luis A. Zárate-Hernández, Jose A. Rodriguez, Eva M. Santos, Julián Cruz-Borbolla

**Affiliations:** Área Académica de Química, Universidad Autónoma del Estado de Hidalgo, Carr. Pachuca-Tulancingo Km. 4.5, Mineral de la Reforma 42184, Mexico

**Keywords:** DFT, organosulfur compounds, acrylamide mitigation

## Abstract

Thermal processing of certain foods implies the formation of acrylamide, which has been proven to provoke adverse effects on human health. Thus, several strategies to mitigate it have been developed. One of them could be the application of organosulfur compounds obtained from natural sources to react with the acrylamide, forming non-toxic adducts. A DFT study of the acrylamide reaction with the organosulfur model compounds L-cysteine and L-glutathione by Michael addition and a free radical pathway complemented by a kinetic study of these model molecules has been applied. The kinetic evaluation results demonstrate that the L-glutathione reaction exhibited a higher rate constant than the other studied compound.

## 1. Introduction

Acrylamide (C_3_H_5_NO) is an unsaturated amide which is soluble in water, colorless, and odorless, and it is formed during food thermal processes by the Maillard reactions (browning reactions) through Strecker degradation between an amino acid (commonly asparagine) and carbonyl or dicarbonyl compounds from reducing sugars. Acrylamide (ACR) can be found in biscuits, bread, breakfast cereals, coffee, and especially fried products (e.g., french fries and chips) [1,2,3]. Apart from undesirable browning, acrylamide has a neurotoxic effect and has been classified as a probable human carcinogen (Group 2A) by the International Agency for Research on Cancer [1]. Therefore, different strategies have been developed to reduce the acrylamide content in the food according to the limits established by EU regulation No. 2017/2158 [4].

The α,β-unsaturated compounds (including acrylamide) can react with nucleophiles such as thiol compounds in a conjugate addition (Michael addition) [5] (Figure 1). In this type of conjugation, the addition is 1,4, in which a nucleophilic attack occurs on the β-carbon of the α,β-unsaturated carbonyl, resulting in an intermediate enolate with a negative charge that generates the Michael adduct by C−C, C−N, C−S, C−O, or other C−X bonds. In the thiol-Michael addition reaction pathway catalyzed by a Lewis base, the reaction kinetics as well as the yield of the final product will depend on factors such as the strength and concentration of the base catalyst, the thiol pKa value, the steric accessibility of the thiol, the nature of the electron-withdrawing group coupled to the C=C bond, the polarity, and the pH of the solvent. All these factors will affect the kinetics of the reaction in solution reactions.

On the other hand, when the thiol-Michael addition reaction is catalyzed by a strong nucleophile, a strong base is generated, producing the intermediate zwitterionic enolate base responsible for deprotonating the thiol. Rapid reaction kinetics are expected because of the high reactivity of thiolate anions [6]. In this sense, the vinyl carbon of acrylamide could react with deprotonated thiol (${RS}^{-}$) to form an adduct [7]. Therefore, one strategy to mitigate acrylamide could be the formation of adducts by a Michael addition reaction with antioxidants such as organosulfur compounds from natural sources. Moreover, it has been described that thiols such as glutathione can competitively react with glucose against amino acids, causing a decrease in the content of acrylamide [8].

Regarding the inhibitory pathways, the reported information is scarce, since the Michael adduct formation is responsible for the inhibition, but the stabilization (second step) can follow two different pathways [9]. In the present study, a computational approximation of a possible chemical pathway in the reaction between acrylamide and the model molecules—L-cysteine (L-Cys) and L-glutathione (L-GSH)—via the free radical pathway and the ionic pathway is proposed. This work has been experimentally supported with the aim of evaluating the use of organosulfur compounds in acrylamide mitigation.

## 2. Results and Discussion

### 2.1. Kinetic Evaluation

The reactions among the ACR and model molecules (L-Cys and L-GSH) were evaluated to understand the reaction mechanism in ACR mitigation. As a first step, these reactions were analyzed experimentally at different temperatures and a pH of 7.3 in order to observe their behavior (Figure 2). The data were adjusted to a pseudo-first-order reaction (R${}^{2}$ > 0.9542), and similar performance has been described with respect to ACR in this type of reaction [10]. From this information, the reaction rate constants were calculated, obtaining higher reaction constants with L-GSH than L-Cys (Table 1). The pseudo-first-order rate constants were affected by the pH level because reactions between ACR and thiols improve when the pH increases above 7.0 [10,11]. In addition, the activation energy (Ea), calculated by the Arrhenius equation, was 62.19 and 30.05 $\mathrm{kJ}\cdot {\mathrm{mol}}^{-1}$ for the L-Cys-ACR and L-GSH-ACR reactions, respectively.

For the reaction to occur, the thiol compounds must necessarily follow the proposed two-step mechanism: (1)$$R-SH+H_{2}C=CHCONH_{2}⟶RSH-H_{2}C-CHCONH_{2}⟶RS-H_{2}C-CH_{2}CONH_{2}.$$

### 2.2. Reactivity Analysis

Four global reactivity descriptors of ACR, L-Cys, and L-GSH were calculated: the ionization potential (*I*), electronic affinity (*A*), hardness (η), and chemical potential (μ). Table 2 shows that the compound with the lowest value of *I* was L-GSH, meaning that this compound needed less energy to lose an electron (9.2 eV), followed by L-Cys at 9.4 eV. On the contrary, ACR presented the highest value of *A*, implying that this compound possessed a good chance to accept an electron with −0.8 eV. These results suggest that ACR in the proposed reaction should be an electronic acceptor, and the L-Cys and L-GSH should be the electronic donors. According to Pearson’s theory [12], the reactants need to have similar values of η in order to react with each other. In this case, the results of these compounds were around 5.0–5.4 eV, being quite similar to each other with a difference of 0.1–0.3 eV between the reactants.

The local reactivity was analyzed through the molecular orbitals, as well as the Fukui functions $f^{+}$, $f^{-}$, and $f^{0}$. Figure 3 shows the frontier orbital plots of the reactants involved in studied reactions. The most interesting orbitals were the ACR LUMO located along the delocalized π orbitals or vinyl group and the L-Cys and L-GSH HOMO orbitals located on the sulfur atom of the thiol group. These results are consistent with the proposed mechanism of an initial nucleophilic attack of sulfur on the vinyl group, as in other similar cases [13]. On the other hand, the Fukui functions allowed us to identify susceptible local areas to electrophilic ($f^{-}$), nucleophilic ($f^{+}$), and free radical ($f^{0}$) attacks. In Figure 4, the Fukui functions are displayed in green and blue colors for the positive and negative parts of the function, respectively. In the case of ACR, most of the Fukui function $f^{-}$ was over the oxygen atom, indicating that oxygen is susceptible to electrophilic attacks, but in the L-Cys and L-GSH molecules, the Fukui surfaces were dispersed over the molecule, and therefore a specific susceptible area for nucleophilic attacks was not visualized. On the other hand, the $f^{+}$ function for ACR was located over the β-carbon, while the Fukui functions $f^{-}$ for L-Cys and L-GSH were over the sulfur, suggesting the ability of these molecules to react as a nucleophile with β-carbon atoms from ACR to the adduct. The Fukui functions $f^{0}$ showed the surface located on the sulfur atoms from L-Cys and L-GSH, making them susceptible to free radical attacks, while in ACR, the Fukui function indicated that the free radicals were dispersed around the molecule.

### 2.3. Proposed Mechanism

This analysis was centered on the nature of S−H bond breaks, followed by the formation of Michael adducts [10] with stabilization by the migration of the hydrogen from sulfur to α-carbon. This step of the mechanism was analyzed by two pathways (radical and ionic) following the approximation of the transition state with the isolated species in order to find out the electronic nature of the hydrogen migration and understand the kinetics in the formation of Michael adducts. The energy results of the reactants, products, and both transition states (TSs) of radical and ionic obtained are reported in Table 3 for the reaction between ACR and L-Cys (Rxn-Cys) and for the reaction of ACR with L-GSH (Rxn-GSH). The total energy was 0 K (Δ*E*), the enthalpy (Δ*H*) was 298 K, and the free Gibbs energy (Δ*G*) at 298 K for Rxn-GSH was 69.0 $\mathrm{kcal}\cdot {\mathrm{mol}}^{-1}$, lower than the 75 $\mathrm{kcal}\cdot {\mathrm{mol}}^{-1}$ obtained for Rxn-Cys. The results with PBE, implicit solvent inclusion, and zwitterionic structures were very close, showing the same trends.

Both reactions were exothermic at 298 K, since the ΔH values presented were −18.8
$\mathrm{kcal}\cdot {\mathrm{mol}}^{-1}$ for Rxn-Cys and −28.5
$\mathrm{kcal}\cdot {\mathrm{mol}}^{-1}$ for Rxn-GSH. There were also exergonic reactions at 298 K with ΔG values of −8.3
$\mathrm{kcal}\cdot {\mathrm{mol}}^{-1}$ for Rxn-Cys and −9.5 for Rxn-GSH. Figure 5 shows the graph of a singly occupied molecular orbital (SOMO) for L-Cys and L-GSH adducts in free radical form. In both cases, the localization of the SOMO was over the atoms involved in the mechanism, sulfur, and α-carbon, especially with an isovalue of 0.10. However, when the graphs with a more restrictive isovalue were analyzed (0.15), the L-GSH adduct radical SOMO was located over the α-carbon but not in the sulfur one. Nevertheless, the orbital distribution was more homogeneous and equivalent to the sulfur and α-carbon in the L-Cys adduct radical. This might explain the differences found in the kinetics for both reactions, and we can relate the SOMO distribution in the case of the L-GSH adduct to being preferentially on the α-carbon, favoring the stability of a homolytic pathway.

According to the above discussion, the formation of free radicals in the intermolecular migration of hydrogen in a homolytic pathway is the preferred pathway. This pathway would explain a faster reaction of ACR with L-GSH than with L-Cys, agreeing with the obtained experimental results.

## 3. Materials and Methods

### 3.1. Reagents

L-cysteine (L-Cys), L-glutathione (L-GSH), 5,5${}^{\prime }$-dithiobis-(2-nitro-benzoic acid) (DTNB), acrylamide (ACR), and sodium phosphate monobasic monohydrate were purchased from Sigma Aldrich (St. Louis, MO, USA), and NaOH was purchased from Meyer (Mexico City, Mexico). Solutions were prepared with deionized water (Milli-Q system) with a resistance of ≥18.2 MΩcm (Millipore Bedfore, MA, USA).

### 3.2. Kinetic Evaluation

The experimental kinetic evaluation consisted of two different reactions: (1) L-Cys ($6.4\times 10^{-4}$ M, 25.0 mL) and (2) L-GSH ($6.4\times 10^{-4}$ M, 25.0 mL) with ACR ($1.054\times 10^{-4}$ M, 25.0 mL). The reactions were performed at 70.0, 80.0, and 90.0 °C in a reflux system for 60 min. Solutions were prepared in a phosphate buffer solution (PBS) (60.0 mM, pH 7.3). The aliquots taken from the reactions were 0.1 mL, and every aliquot was mixed with 4.0 mL of PBS (60.0 mM, pH 7.3) and 0.5 mL of DTNB (5.0 mM) prepared in PBS (60.0 mM, pH 7.3)). The mixture was analyzed at a wavelength of 412 nm in a Perkin Elmer Lambda 40 spectrometer (Waltham, MA, USA) using Perkin Elmer WinLab software. The pseudo-first-order rate constant ($k_{obs}$) was calculated according to [11]:(2)$$\ln\left(\frac{\left[\mathrm{SH}\right]}{{\left[\mathrm{SH}\right]}_{0}}\right)=-k_{obs}t,$$
where [SH] is the concentration of thiols at a certain time and [SH]${}_{0}$ is the initial concentration of thiols.

### 3.3. DFT Studies

The calculation of the electronic structure of the reactants was carried out using Gaussian16 software [14] with the functional wB97XD [15] and PBE [16] and the basis set 6-311+G** [17,18,19,20] for both functionals. The solvent water was included in the PBE calculations with the model SMD [21]. The ionization potential (*I*) [22] was calculated with the equation
(3)$$I=E^{N-1}-E^{N},$$

For the electronic affinity (*A*), the equation used was
(4)$$A=E^{N}-E^{N+1},$$
where $E^{N-1}$, $E^{N}$, and $E^{N+1}$ are the one-less-electron, neutral, and one-more-electron energies for each compound. With these values, the hardness (η) [23] was calculated with the following equation:(5)$$\eta =\frac{I-A}{2}$$

The chemical potential (μ) [23] was calculated with the following equation:(6)$$\mu =-\frac{I+A}{2}.$$

The structures, energies, and frequencies of the reactants, the products, and the two possible transition states were optimized and calculated, with the first by free radicals and the second by the formation of charged species (ionic species). These transition states were approximated by calculating the isolated species. The products and ions were calculated with a closed shell (singlet multiplicity), and the free radical transition states were calculated using an open shell with doublet multiplicity. In order to be comparative values of energy, the stabilization of the adduct was calculated, with isolated free radicals and ionic species formed after the adduct formation.

## 4. Conclusions

The awareness about ACR has increased in recent years due to its adverse effects on human health. Thus, different strategies have been developed to mitigate its content in food products. One of them could be the application of antioxidants such as organosulfur compounds to form non-toxic adducts. This project evaluated the viability of a homolytic pathway or ionic pathway in the stabilization of Michael adducts for the reaction between ACR and thiols (L-Cys and L-GSH). The results obtained in the experimental evaluations were related to the theoretical analysis to demonstrate that the reactivity of L-GSH and the reaction rate constant of L-GSH were superior to the L-Cys results.

According to the results, it seems that the homolytic pathway was the stablest mechanism in the reaction of thiol from L-GHS with acrylamide as ΔG in the formation of radical species was more stable, with values of 75 kcal·mol${}^{-1}$ and 69 kcal·mol${}^{-1}$ for interaction with L-Cys and L-GSH, respectively, than the values of the ionic species, with values of 351.2 kcal·mol${}^{-1}$ and 296.1 kcal·mol${}^{-1}$ for interaction with L-Cys and L-GSH, respectively, suggesting stabilization of Michael adducts via free radicals is a better option, and this reaction mechanism would explain the higher reactivity and superior reaction rate constant experimentally obtained for L-GSH compared with that for L-Cys. Finally, these results also allow a possible extrapolation of the thiol compounds present in natural sources.

## Figures and Tables

**Figure 1 molecules-27-08220-f001:**
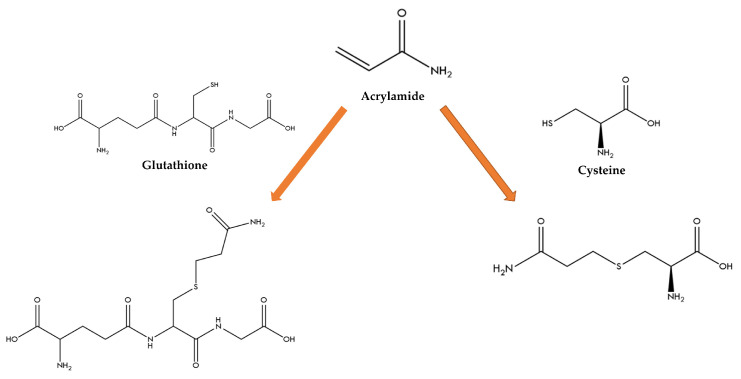
Scheme of the Michael addition reaction for ACR with L-GSH and L-Cys.

**Figure 2 molecules-27-08220-f002:**
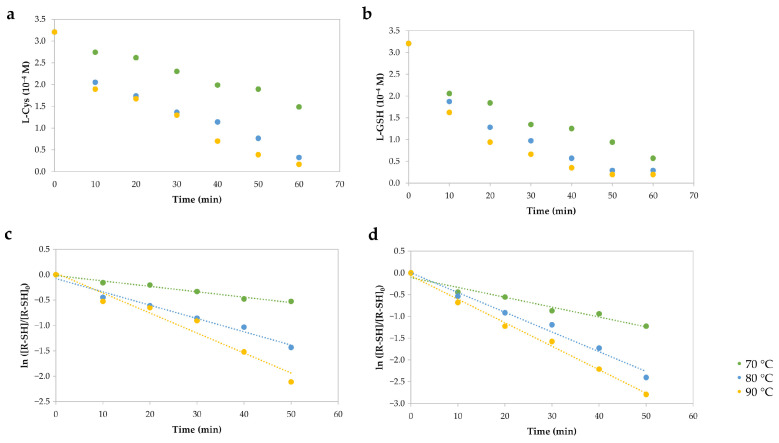
Reaction kinetics parameters of (**a**) L-Cys and (**b**) L-GSH with ACR at pH 7.3 and different temperatures. Pseudo-first-order rate constants ($k_{ObS}$) for (**c**) L-Cys and (**d**) L-GSH were determined from plots of $\ln\left(\left[R-\mathrm{SH}\right]/{\left[R-\mathrm{SH}\right]}_{0}\right)$ vs. time.

**Figure 3 molecules-27-08220-f003:**
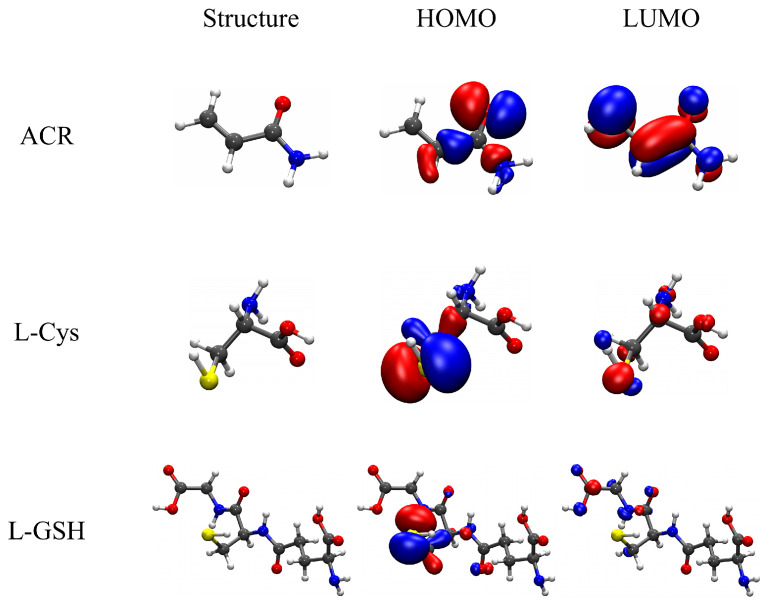
Frontier molecular orbitals of the calculated and studied compounds.

**Figure 4 molecules-27-08220-f004:**
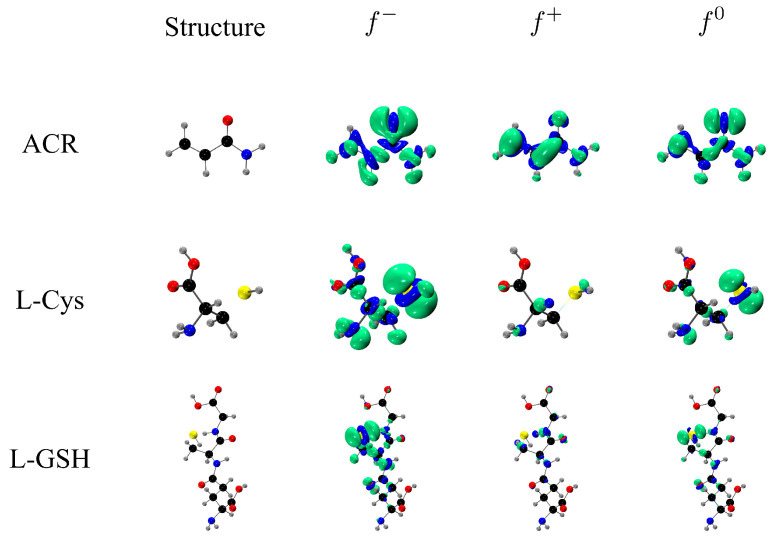
Fukui functions calculated for the reactants.

**Figure 5 molecules-27-08220-f005:**
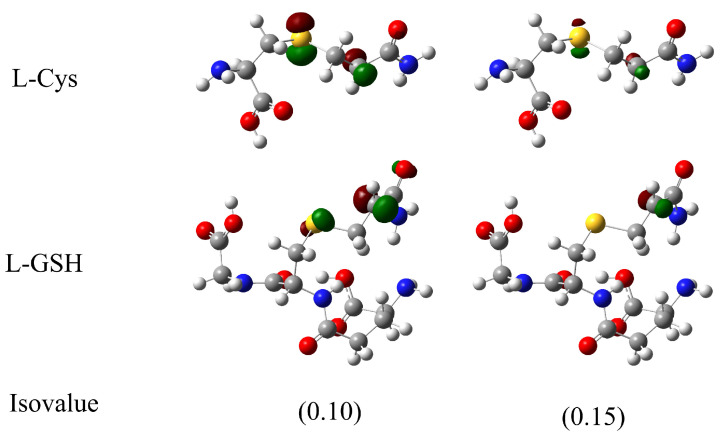
SOMO orbital graphs for modeled transition state free radicals with isovalues of 0.10 and 0.15.

**Table 1 molecules-27-08220-t001:** Pseudo-first-order rate constants (10${}^{-4}$ s${}^{-1}$) for the reactions of $6.4\times 10^{-4}$ M L-Cys and L-GSH with $1.054\times 10^{-4}$ M ACR at different temperatures.

Temperature (ºC)	L-Cys	L-GSH
70	1.98±0.37	4.33±0.11
80	4.37±0.21	5.84±0.67
90	6.57±0.26	7.73±0.46

**Table 2 molecules-27-08220-t002:** Global DFT reactivity descriptors in eV.

Compound	*I*	*A*	η	μ
ACR	9.9	−0.8	5.3	−4.5
L-GSH	9.2	−0.9	5.0	−4.1
L-Cys	9.4	−1.3	5.4	−4.1

**Table 3 molecules-27-08220-t003:** Thermodynamics of reactions of ACR with thiols L-Cys and L-GSH in $\mathrm{kcal}\cdot {\mathrm{mol}}^{-1}$. R = radical, Ion = ionic, and Z = zwitterion.

TS	Thiols	Method	ΔE	ΔH	ΔG
R	L-Cys	wB97XD	75.3	72.0	75.0
Ion	L-Cys	wB97XD	351.4	347.4	351.2
R	L-Cys	PBE (SMD)	70.0	67.1	72.5
Ion	L-Cys	PBE (SMD)	159.1	155.4	160.0
R-Z	L-Cys	PBE (SMD)	70.5	67.5	71.7
Ion-Z	L-Cys	PBE (SMD)	152.9	150.3	155.9
R	L-GSH	wB97XD	63.5	60.2	69.0
Ion	L-GSH	wB97XD	290.9	287.8	296.1
R	L-GSH	PBE (SMD)	68.2	64.9	71.7
Ion	L-GSH	PBE (SMD)	160.8	157.3	162.5
R-Z	L-GSH	PBE (SMD)	68.4	65.0	70.5
Ion-Z	L-GSH	PBE (SMD)	160.8	157.1	161.7

## Data Availability

Not applicable.

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
