# Peer review of "A DFT Study of the Reaction of Acrylamide with L-Cysteine and L-Glutathione"

_molecules, 2022, doi:10.3390/molecules27238220_

Round 1
Reviewer 1 Report
In this study the Authors have used combined theoretical an experimental methods to describe the possible mechanism of the reaction between the acrylamide and gluthathione. The msnuscript is clear and concise, however it also requires revision.
My major comment is that since the Authors have studied the kinetics at various temperatures, which I strongly support, they should have calculated the activation energy (Ea). This can be done in a few minutes using Arrhenius equation. And, since the Authors have used the DFT to support their experimental findings, should compare the already determined (calculated) theoretical Ea with the experimental one (above, I have described how to obtain it). By doing such comparison the Authors would be able to confirm their hypothesis. I am aware that it requires a few additional calculations, but in my opinion this is crucial to increase the scientific value of the manuscript.
Other major issues:
Figure 1, why the similar figure has not been created for L-Cys?
DFT calculations: have the authors used and implicit solvent model? PCM? And what about the dispersion corrections (i.e. D3)? Also, it is not stated directly, how many negative frequencies have been obtained (I hope that zero for optimized structures and one for TS).
From the presented structures it can be seen that the Authors have modeled neutral compounds. However, at neutral pH, both the L-Cys and L-GSH exist as zwitterions. Why the Authors have not decided to model the zwitterions?
Minor issues:
Line 11, please replace „aqueous solutions” with “water”
Line 16, it should be “french fries”
Line 17, replace “neurotoxicity” with “neurotoxic”
Line 19, please remove “recently”
Lines 21-23, a Figure should be created to show the scheme of the reaction discussed in those lines, to increase the clarity
Lines 49 and 51 “were evaluated” was used twice in a row
Line 58, there is a grammar mistake in this sentence
Reviewer 2 Report
The manuscript under consideration presents both theoretical and experimental studies of the reaction of ACR with L-Cys and L-GSH. My background is in density functional theory (DFT). Therefore, I can only evaluate the DFT part of the manuscript.
The main goal of the manuscript is to study whether the homolytic or ionic pathway is possible in the reaction between ACR and L-Cys/L-GSH. To understand that the authors presented four descriptors obtained from density functional theory. In addition, three types of energy barriers corresponding to the homolytic and ionic pathways are calculated as well.
I cannot recommend the current manuscript for publication. Frist, it is difficult to see how the descriptors defined in Sec. 3.3 could be used to understand the homolytic and ionic pathways. Second, the energy barriers presented in Table 3 and 4 are not defined in the manuscript. I can only assume that E, H and G are the total energy, the Helmholtz free energy and the Gibbs free energy. As I understand it, density functional theory calculations are usually carried out at the zero temperature. Are the free energy barriers in Table 3 and 4 calculated at zero temperature? If so, what’s the impact of non-zero temperature?
A few minor comments:
1). In line 64, “loose” should be “lose”
2) Why is WB97XD chosen as the exchange correlation functional?
3) What do green and brown represent in Fig. 4?
Round 2
Reviewer 1 Report
The Authors have made the requested corrections. Current version can be accepted for publication.
Reviewer 2 Report
The authors have addressed my questions. However, I should note that my area of interest is density functional theory. I cannot judge the quality and the validity of the experimental results presented in the manuscript.